# A PERSPECTIVE OF IMPROPER DYNAMICS ON OFFLINE MODEL-BASED PLANNING

## ABSTRACT

By learning the dynamics model, estimating, and planning on the latent state, MuZero and its variants perform well in complex environments. However, the performance of these algorithms require an accurate dynamics model and prediction model, which may be difficult in offline reinforcement learning since the lack of interactions with the environment. Recent works attempt to use one-step rollouts to reduce the cumulative error of rollout caused by an inaccurate dynamics model. We argue that the planning issues of MuZero-type methods are mainly caused by inaccurate models. To address this issue, we propose a robust method, **C**onstrained **O**ffline **M**odel-based **P**lanning (COMP), for training dynamics or prediction models more smoothly. COMP introduces a kind of specifically designed noise to the latent state, aiming to align the value and dynamics of these states with those of states not perturbed. Our method can be combined with MuZero and its derived algorithms to improve planning performance in offline settings. Experiments show that our proposed method achieved notable performance in most Atari game tasks on RL Unplugged benchmark.

## 1 INTRODUCTION

Offline reinforcement learning, also known as batch reinforcement learning (Riedmiller (2005); Lange et al. (2012)), is a variant of reinforcement learning in which an agent learns from a fixed dataset without interacting with the environment, with the goal of achieving the maximum reward. Offline reinforcement learning is a promising solution for applying reinforcement learning on a large scale to practical fields (Jaques et al. (2019); Ebert et al. (2018); Kalashnikov et al. (2018); Kahn et al. (2021)), especially those with high costs associated with interacting with the environment, such as autonomous driving (Kiran et al. (2021)) and healthcare (Liu et al. (2020)). One challenge of offline reinforcement learning is that the agent may overestimate the value of state-action pairs outside the dataset or select out-of-distribution (OOD) actions, without the ability to interact with the environment to correct its error estimation. Like other subfields of reinforcement learning, offline reinforcement learning can be divided into two categories to address the challenges it presents: model-free (calculate a conservative or high-confidence value to constrain the action, e.g. Kumar et al. (2020), Kostrikov et al. (2021)) and model-based (generate a high credit value for the environment's state transition, or plan more carefully, e.g.Kidambi et al. (2020), Schrittwieser et al. (2021)). In model-free reinforcement learning, an agent learns to generate actions without explicitly modeling the environment. On the other hand, model-based reinforcement learning helps agents generate plans by learning the dynamics of the environment, achieving higher data efficiency than model-free methods. Researchers hope that the characteristics of model-based reinforcement learning (RL) will provide an advantage in addressing the challenges of offline settings. Among the various model-based algorithms, our focus is on the offline method of MuZero because, as a value-based, model-based planning algorithm, it outperforms other model-based algorithms in complex environments, such as Atari games (Bellemare et al. (2013)).

In previous research, MuZero Unplugged (Schrittwieser et al. (2021)) and ROSMO (Liu et al. (2022)) are two offline reinforcement learning algorithms that generate actions by planning with a learned dynamics model. MuZero Unplugged is a simple extension of the MuZero algorithm that learns a dynamics model and uses Monte Carlo Tree Search (MCTS) (Coulom (2006), Kocsis & Szepesvári (2006)) for planning. ROSMO is an improvement over MuZero Unplugged in offline RL. The authors of the paper noted that MuZero Unplugged may perform poorly in situations such

as learning with limited data coverage and using improperly parameterized models given the offline data, among others. ROSMO uses one-step look-ahead to mitigate these issues and, after incorporating an advantage filtering regularization term, achieves better performance than MuZero Unplugged. However, while ROSMO avoids the cumulative error of rollout error by using one-step rollouts, it has not attempted to solve the problem of rollout error itself. In fact, the error in state translation introduced by an improper dynamics model still influences the agent. This error may mislead planning by imaging an erroneous, out-of-distribution latent state.

The planning process of above-mentioned algorithms in evaluation requires the model to output the probability of actions, predicting the value, and translating the latent state for the most probable action. According to the algorithm, this process repeatedly performs various steps for the action sequence, allowing the model to obtain the value of actions and select the action with the highest value. By examining the plan process of these algorithms, we find that even though ROSMO reduces the cumulative error, if the rollout error occurs in a one-step rollout, the plan function will be affected. For value-based planning methods in offline settings, the poor performance of the dynamics model will have an great impact on the prediction model. If the dynamics model imagines a latent state that the prediction model has never visited and the prediction may overestimate this latent state, similar to what is faced by model-free methods. Ultimately, this will impact planning in the evaluation phase, even if there is a powerful planning function. This is because, in the planning phase, the selection of an action depends on the value of that action. This may be misled by an erroneous value associated with a latent state that has never been seen before. To address the challenges faced by value-based planning algorithms in an offline setting, we have drawn inspiration from the DrQ (Kostrikov et al. (2020); Yarats et al. (2021)). This has led us to design a **C**onstrained **O**ffline **M**odel-based **P**lanning (COMP) algorithm. We add noise to the latent state to perform robust training. By robustly training the dynamics or prediction model, the algorithm will perform more smoothly in rollouts. In addition, as discussed in Zheng et al. (2023), adding noise can construct a local Lipschitz condition to learn a better model. Our experiments in Atari games show that our algorithm has a smaller noise loss than ROSMO and its one-step rollout method. In this paper, we focus on adding our method to the one-step rollout proposed by ROSMO. The effectiveness of our method on MuZero Unplugged has been proven by some experiments. We compare our algorithm with state-of-the-art methods such as MuZero Unplugged (Schrittwieser et al. (2021)), ROSMO (Liu et al. (2022)), and Critic Regularized Regression (Wang et al. (2020)) in the large-scale RL Unplugged Atari benchmark (Gülçehre et al. (2020)). Our algorithm achieved a notable score in several tasks. Besides, ablations provide further analysis about the sensitivity of hyper-parameters and the effectiveness of the noise design.

## 2 BACKGROUND

### 2.1 OFFLINE REINFORCEMENT LEARNING

The goal of reinforcement learning is to train a policy to maximize the expected cumulative reward, which experts usually set. Many RL algorithms are considered in the Markov decision process (MDP), which contains a tuple $(\mathcal{S}, \mathcal{A}, T, p_0, r, \gamma)$. $\mathcal{S}, \mathcal{A}$ represent state spaces of environment and action spaces of policy, $T(\mathbf{s}' \mid \mathbf{s}, \mathbf{a})$ is environment dynamics and tell us the transition probability of state s' when policy uses action a in state s. $p_0$ defines the initial state distirubtion. $r(\mathbf{s}, \mathbf{a})$ represents reward function, and $\gamma \in (0, 1)$ is the discount factor. Typically, model-based reinforcement learning algorithms need to learn the environment dynamics $T(\mathbf{s}' \mid \mathbf{s}, \mathbf{a})$ to generate a new state $\mathbf{s}'$ and reward $r(\mathbf{s}, \mathbf{a})$ for a specific state-action pair. $\pi_\beta(\mathbf{a} \mid \mathbf{s})$ represents behavior policy which use to collect dataset D, $d^{\pi_\beta}(\mathbf{s})$ is discount marginal state-distribution of $\pi_\beta(\mathbf{a} \mid \mathbf{s})$. The goal of the policy is to maximize returns:

$$\pi^* = \arg\max_\pi \mathbb{E}_\pi \left[ \sum_{t=0}^{\infty} \gamma^t r(s_t, a_t) \mid s_0 \sim p_0(\cdot), a_t \sim \pi(\cdot \mid s_t), s_{t+1} \sim p(\cdot \mid s_t, a_t) \right] \quad (1)$$

### 2.2 OFFLINE VALUE-BASED PLANNING

Model-based RL usually makes effective use of data by learning an environment dynamics and reward model (although this is not required for all model-based RL algorithms), making it promising for solving the offline RL problem. Unlike most model-based methods, MuZero learns a latent state from observations without reconstructing the observation and learns a dynamics model for planning.

In this way, the MuZero algorithm has notable performance in complex visual environments. For MuZero and its improved algorithm, they generate actions by planning with model rollouts and evaluating the value of every potential action for the current state. Therefore, we describe a general algorithmic framework extended from MuZero as value-based planning.

Given a trajectory $\tau_i = \{o_1, a_1, r_1, ..., o_{T_i}, a_{T_i}, r_{T_i}\} \in D$ and any time step $t \in [1, T_i]$, the model encodes the observation of $t$ into a latent state via the representation model $h_\theta : s_t^0 = h_\theta(o_t)$. Then, the algorithm uses the dynamics model $g_\theta$ to obtain an imagined next state in the latent space and a reward: $r_t^{k+1}, s_t^{k+1} = g_\theta(s_t^k, a_{t+k})$, where $k \in [0, K]$ represents the imagination depth of the output (the steps we unroll using the dynamics model $g_\theta$). In addition, the prediction model predicts the policy and value $\pi_t^k, v_t^k = f_\theta(s_t^k)$ conditioned on the latent state. Finally, when the agent tries to generate an action, the prediction function outputs a series of potential actions and selects an action that may have the maximum value in its prediction. This process is called value-based planning.

The above network parameters will be updated by following the loss objective in the training phase,

$$\ell_t(\theta) = \sum_{k=0}^{K} \ell^r \left(r_t^k, r_{t+k}^{\text{env}}\right) + \ell^v \left(v_t^k, z_{t+k}\right) + \ell^\pi \left(\pi_t^k, p_{t+k}\right) + \ell^{\text{reg}}(\theta) \tag{2}$$

where $l^r, l^v, l^\pi$ are loss functions of reward, value and policy respectively. $l^{reg}$ is a regularization term that changes according to the needs of the algorithm. The loss function for the aforementioned objective is cross-entropy, the details of which will be discussed in the Appendix A. $r_{t+k}^{env}$ represents the actual reward from the environment, while $z_{t+k}$ is computed by adding the discounted actual reward to the target value of $t+k+n$ (where $n$ is the number of steps required for value calculation.). $p_{t+k}$ represents the improved action, as decided by the algorithm, selected by planning in the target network. One difference between MuZero and other model-based methods is that it does not require supervised training of the dynamics and representation functions. In other words, the model does not need to reconstruct the predicted next latent state back to the input space. The idea behind this method is the value equivalence principle (Grimm et al. (2020)): if the value of a state-action pair is accurate, the model can improve the policy, regardless of whether the latent state can be accurately reconstructed in the input space.

### 2.3 POLICY IMPROVEMENT METHOD

MuZero Unplugged and ROSMO have different methods of improvement. MuZero Unplugged uses Monte-Carlo Tree Search (MCTS) (Coulom (2006), Kocsis & Szepesvári (2006)) to compute the target value and policy in a plan with a simulation budget of $N$. Starting from the root latent state $s_t^0$, the model simulates a series of trajectories based on MCTS with a budget of $N$. In every simulation, the model selects an action according to the following pUCT (Rosin (2011)) rule:

$$a^k = \arg\max_a \left[Q(s,a) + \pi_{\text{prior}}(s,a) \cdot \frac{\sqrt{\sum_b n(s,b)}}{1 + n(s,a)} \cdot \left(c_1 + \log\left(\frac{\sum_b n(s,b) + c_2 + 1}{c_2}\right)\right)\right] \tag{3}$$

where $n(s,a)$ is the number of times the state-action pair has been visited, $Q(s,a)$ represents the current estimate of the state-action value, $\pi_{prior}(s,a)$ is computed by the prior policy, and $c_1$ and $c_2$ are constants. Then, the model predicts its reward $r_t^l$ and value $\frac{l}{t}$ and reaches a leaf node $s_t^{l+1}$. By repeating this process until the simulation budget is exhausted, the model completes an MCTS and obtains the return $G^k = \sum_{\tau=0}^{l-1-k} \gamma^\tau r^{k+1+\tau} + \gamma^{l-k} v^l$. Then it computes the target policy and value:

$$p_{\text{MCTS}}(a \mid s_t) = \frac{n\left(s_t^0, a\right)^{1/T}}{\sum_b n\left(s_t^0, b\right)^{1/T}}$$

$$z_{\text{MCTS}}(s_t) = \gamma^n \sum \left(\frac{n\left(s_{t+n}^0, a\right)}{\sum_n n\left(s^0, b\right)}\right) Q\left(s_{t+n}^0, a\right) + \sum^{t+n-1} \gamma^{t'-t} r_{t'}^{\text{env}}. \tag{4}$$

On the other hand, ROSMO simulates a one-step process and selects the policy by following the given equation.

$$\ell_{\text{OS}}^\pi = -\mathbf{p}^\top \log \boldsymbol{\pi} \tag{5}$$

where policy target **p** is computed as:

$$p(a \mid s) = \frac{\pi_{\text{prior}}\left(a \mid s\right)\exp\left(\text{adv}_g(s,a)\right)}{Z(s)} \tag{6}$$

where $adv_g = (q_g(s,a) - v(s))$, $Z(s)$ is normalization term.

## 3 METHOD

### 3.1 ACCUMULATED ERROR OF DYNAMICS MODEL

As we discussed above, model-based planning algorithms need model to predict latent states, simulate future probability transitions, and predict the value of latent states to improve policy. In this process, the accuracy of the dynamics or prediction model is important because the critic value of every action will influence the final selection of the agent. When the dynamics model generates an out-of-distribution state, the prediction model will predict an erroneous value for this state. This value may mislead the improvement process to select an improper action by overestimating the state. In MuZero Unplugged, depending on the simulation depth, this rollout error will accumulate because the improper dynamics model will pull the imagined latent state far from the distribution. Although ROSMO uses one-step look-ahead to reduce the accumulation of error, the error still influences the improvement process because it is never solved.

We analyze the improvement process of ROSMO and MuZero Unplugged to understand why action selection errors occur and how to reduce them. We begin this process with ROSMO:

$$p^*(a \mid s) - p(a \mid s) = \frac{\pi_{\text{prior}}\left(a \mid s\right)\exp\left(\text{adv}_g^*(s,a)\right)}{Z(s)} - \frac{\pi_{\text{prior}}\left(a \mid s\right)\exp\left(\text{adv}_g(s,a)\right)}{Z(s)}$$
$$= \pi_{\text{prior}}\frac{\exp\left(\text{adv}_g^*(s,a)\right) - \exp\left(\text{adv}_g(s,a)\right)}{Z(s)} \tag{7}$$

To select the correct action, the model must reduce the error when the dynamics model generates an improper latent state. Because the exponential function is a monotonic function, reducing the difference between $\exp\left(\text{adv}_g^*(s,a)\right)$ and $\exp\left(\text{adv}_g(s,a)\right)$ can be translated as minimizing the difference between $\text{adv}_g^*(s,a)$ and $\text{adv}_g(s,a)$. Then, by introducing the concept of advantage value, we can obtain the following equation:

$$min(\text{adv}_g^*(s,a) - \text{adv}_g(s,a)) = min((Q^*(s,a) - V(s)) - (Q(s,a) - V(s)))$$
$$= min((r(s,a) + V(s'^*) - V(s)) - (r(s,a) + V(s') - V(s))) \tag{8}$$
$$= min(V(s'^* - V(s')) = min(V(s') - V(s' + pw))$$

where $s + pw$ represents the true latent state with a noise component along unit vector $w$ with weight $p$.

Although MuZero Unplugged uses the visit count for action selection during the evaluation phase, the value of state-action pairs is still significant when the model simulates a fixed number of steps. This value can influence the visit count for each action and may ultimately alter policy selection. Similar to ROSMO, we also present the error of MuZero Unplugged when an inadequate dynamics model provides an improper latent state during the simulation phase. The pUCT rule, which controls action selection during simulation, is composed of two parts: the value of state-action pairs and the prior policy probability distribution related to the visit count. In this phase, the value of state-action pairs holds more importance than the visit count, leading us to overlook any errors in the visit count.

$$a^{*k} - a^k = \arg\max_a \left[Q(s,a)^* - Q(s,a)\right]$$
$$= \arg\max_a \left[\sum_{\tau=0}^{l-1-k}\gamma^\tau r^{*k+1+\tau} + \gamma^{l-k}v^{*l} - \sum_{\tau=0}^{l-1-k}\gamma^\tau r^{k+1+\tau} + \gamma^{l-k}v^l\right]$$
$$= \arg\max_a \left[\sum_{\tau=0}^{l-1-k}\gamma^\tau r(s^{k+1+\tau}, a^{k+1+\tau}) + \gamma^{l-k}v(s^l) - \right. \tag{9}$$
$$\left. \sum_{\tau=0}^{l-1-k}\gamma^\tau r(s^{k+1+\tau} + q^{k+1+\tau}w^{k+1+\tau}, a^{k+1+\tau}) + \gamma^{l-k}v(s^l + q^l w^l)\right]$$

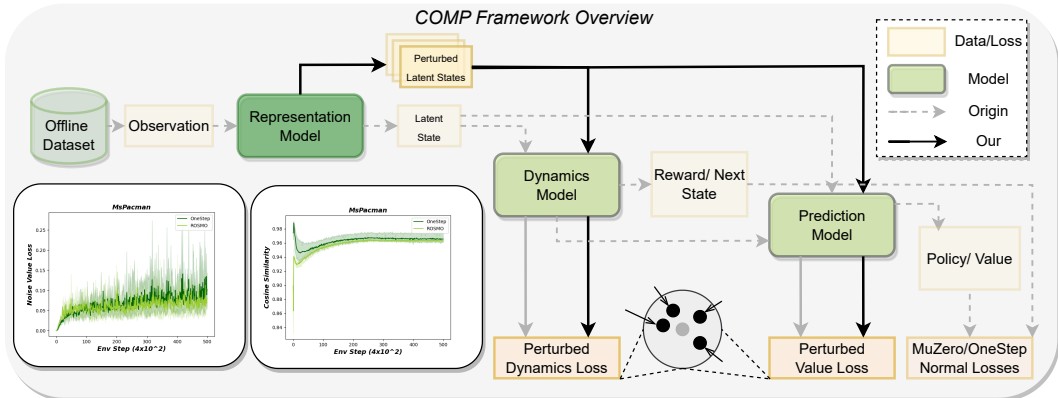

Figure 1: This figure illustrates the framework of our COMP algorithm. Our algorithm utilizes two methods:(1) we introduce noise into the initial latent state and generate a state sequence, then compute the cosine similarity; (2) we introduce noise into all state sequences and generate their values, then compute the difference. The aim of our algorithm is to bring the state with noise closer to the state without noise by using a perturbed loss function. The left side of the figure displays the cosine similarity and difference between the state with noise and the state without noise for both the RMOS and OneStep algorithms. It reveals that previous algorithms have not reduced these values.

By utilizing Equation 8 and Equation 9, we discover that minimizing the error of action selection necessitates minimizing the difference between the value of the state and the value of the noise state, or in other words, minimizing the translation error. Additionally, it's important to note that simulations in MuZero Unplugged can cause the error to spread to the reward of the state-action. Thus, minimizing the discrepancy between the reward of a state and the reward of a noise state can contribute to the selection of the correct action.

## 3.2 ROLLOUT ERROR IN OFFLINE SETTING

Although a poor dynamics model can spread error through simulation and lead the prediction model to generate misleading values, RL agents can correct it through interaction with the environment. In an online setting, the RL algorithm can interact with the environment to explore more states to correct this error. However, in an offline setting, the RL algorithm trains with a fixed dataset. The prediction model cannot correct the errors introduced by improper latent states without interaction. Therefore, these errors will accumulate during the training phase. During the evaluation phase, these errors can influence the model's improvement process and lead to the selection of suboptimal actions. Consequently, both ROSMO and MuZero Unplugged continue to be affected by this error.

We calculate the difference between the value of the state and the value of the noise state, as well as the cosine similarity between these states. The noise of a state follows a distribution where the mean is the state itself and the standard deviation is a function of the state's weight and a base value:$pw \sim \mathcal{N}(0, s * w + b)$. For both cosine similarity and state value difference, we employ two methods to introduce noise into the latent state. When calculating the state value difference, we add noise to all five steps of the unroll process, prompting the prediction model to generate a noisy value. Conversely, for cosine similarity, we introduce noise into the initial state and then allow the dynamics model to roll out and generate a noisy state sequence. As shown in Figure 1 for the MsPacman task, we observe that the ROSMO, OneStep prediction model, and dynamics model are all affected by noise. Furthermore, due to the significant numerical differences within the latent state itself and the number-insensitive nature of cosine similarity, the cosine similarity between these states may be greater than their actual similarity.

### 3.3 FRAMEWORK AND METHOD

To minimize the error caused by improper latent state translation, we are focusing on constraining the difference between the state and the noise state. This idea is inspired by the DrQ algorithm, which uses image augmentation to enhance performance. Instead of augmenting the dataset in observation as done in DrQ, we have decided to add noise in the latent state layer. As shown in Figure 1, these model-based planning methods operate within the latent state layer. The improper translation also occurs in the latent state layer. Therefore, by adding noise to the latent state and computing the difference between them, we can constrain the policy.

Our algorithm only modifies the training phase of OneStep or MuZero Unplugged. As shown in Figure 1, after the representation model generates the latent state, we add noise using different methods according to the constraining aim. Then, we use different objective losses as regular terms to update the network. Through analysis and experimentation, we have used two methods to add noise at different positions. In this paper, we employ two types of noise: (1) sampling from a normal distribution with weights, and (2) sampling from a uniform distribution within the range that is controlled by a parameter, and adding it to the state on a one-to-one basis with weights. Although the error of the state may have a complex form, we opt for these simple methods.

#### 3.3.1 CONSTRAINED DYNAMICS

One of the two methods involves adding noise to constrain the dynamics model. Inspired by EfficientZero (Ye et al. (2021)), we measure the difference between states using cosine similarity. EfficientZero computes the cosine similarity between the state generated by the dynamics model and the state from the representation model. However, unlike EfficientZero, which constrains the image latent state to closely resemble the real latent state, we constrain the image state sequence starting from a state with noise to closely resemble the state sequence without noise.

As shown in Figure 1, we introduce noise into the initial latent state, which is generated by the representation model. The dynamics model then rolls out 'n' steps to obtain a state sequence. We compute the cosine similarity between this state sequence and another state sequence generated through a normal process. Although the error in the latent state is more complex, our experiments demonstrate that our simple method can enhance the performance of the algorithm.

#### 3.3.2 CONSTRAINED VALUE

Another method to introduce noise into the model is by constraining the values generated by the prediction model. The goal of this method is to smooth the state values to generate appropriate values for out-of-distribution states. Moreover, both experimental and theoretical analyses of Zheng et al. (2023) demonstrate that a robust regularization term achieves local Lipschitz conditions. This can enhance performance and train a more accurate prediction model.

Unlike the dynamics constraint method, which only adds noise to the initial latent state, our method introduces noise to all latent states and generates their values. We then compute the difference between these states and their counterparts without noise to smooth the values. As discussed in Section 3.3.1, our experiments demonstrate that our method improves the model's performance, even though it may be too simplistic to simulate complex conditions of improper state transitions.

## 4 EXPERIMENT

In this section, we first compare the cosine similarity and the difference between the state and noise state of our algorithm and other methods to validate the effectiveness of our method in reducing errors introduced by improper state translation (Section 4.1). Next, we compare our algorithm with MuZero Unplugged, ROSMO, and OneStep in the RL Unplugged Atari benchmark. Detailed results will be provided in the Appendix C. Finally, in Section 4.3, we analyze the impact of position, number, and weight of noise through experimental analysis.

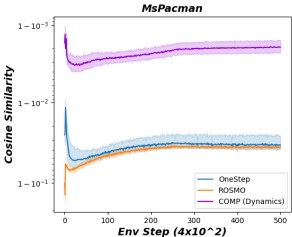 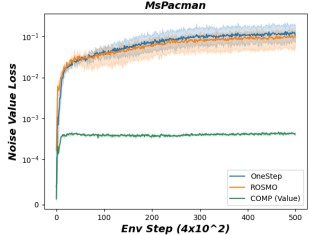 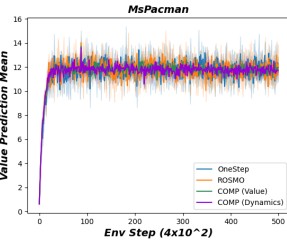

Figure 2: In these figures, we compare our algorithm with ROSMO and OneStep in Atari games. We plot the cosine similarity and mean squared error (MSE) difference between the state and the noise state. The left and middle sections of this figure illustrate that our algorithm constrains the value of counterpart criteria. Additionally, we demonstrate the prediction of in-dataset values to substantiate that our algorithm has a small impact on the in-dataset state value.

### 4.1 HYPOTHESIS VERIFICATION

As demonstrated in Section 3.2, the previous offline model-based planning algorithm encounters errors due to improper state translation. Therefore, when given a state with noise, the algorithm exhibits significant errors in value and state translation. In contrast, our algorithm improves performance by constraining the value difference or the cosine similarity between perturbed states.

To validate the effectiveness of our algorithm, we conducted a comparative analysis of the corresponding results under the two aforementioned metrics between our algorithm, ROSMO, and OneStep. Figure 2 illustrates that these two methods achieve their respective aims independently, without any mutual interference. Furthermore, as depicted on the right side of Figure 2, our algorithm does not alter the values and rewards of in-dataset states. This is because all values of the dataset's states fluctuate within a similar range. As a result, our algorithm enhances smoothness and has minimal negative impact on the original method.

### 4.2 MAIN EXPERIMENT

After verifying the effectiveness of our algorithm in constraining values and dynamics, we compared our method with other offline reinforcement learning algorithms the Atari Games Benchmark provided by RL Unplugged (Gülçehre et al. (2020)). In this section, we compare our algorithm with ROSMO, OneStep, and MuZero Unplugged respectively in three Atari games. Additional comparison results will be presented in the Appendix B.

**Network.** For a fair comparison, we run our algorithm, ROSMO, OneStep, and MuZero Unplugged with a similar network. We use the implementation of ROSMO and run other methods by adding MCTS, a simulation function, or other regularization terms. In addition, we add our constraint regularization term to MuZero Unplugged and OneStep respectively to demonstrate the effectiveness of our algorithm in one-step rollout and multi-step rollout. Notably, the ROSMO algorithm is the OneStep method with an added behavior regularization term.

**Result.** First, we compare our algorithm with ROSMO and OneStep to verify that our algorithm can improve performance in the one-step method. As Figure 3 shows, although the ROSMO method has a faster convergence speed, compared to its biased method OneStep, our algorithm achieves better performance, particularly in the Amidar game. In addition, the main factor of the convergence speed of ROSMO is the behavior regularization term, which can combine with our regularization term.

Then, we compare our algorithm with MuZero Unplugged by adding a regularization term. As discussed in Section 3.2, in MCTS, the reward will also generate an error, so we constrain both the value and the reward in this experiment. Figure 4 demonstrates that our algorithm surpasses MuZero Unplugged. Multi-step rollout simulations introduce a more complex error in state translation. Therefore, a more sophisticated perturbation process will yield better effectiveness. For a more in-depth study of noise addition in MuZero Unplugged, we have put it into our future research.

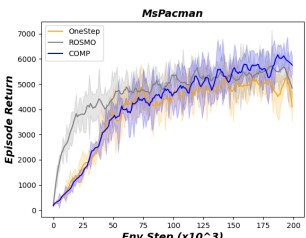 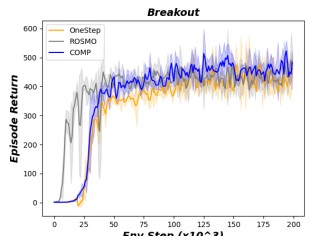 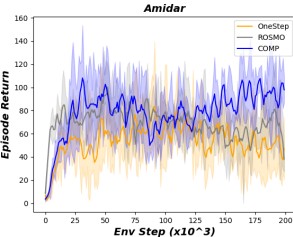

Figure 3: In these figures, we compare our algorithm with ROSMO and OneStep in Atari games. We plot the IQM score for the MsPacman, Amidar, and Breakout tasks. Notably, in these comparative experiments, we add our regularization term to OneStep to ensure fairness.

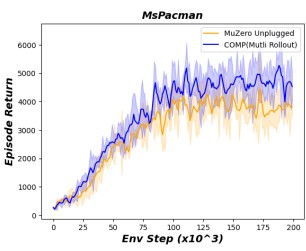 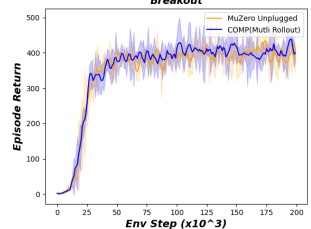 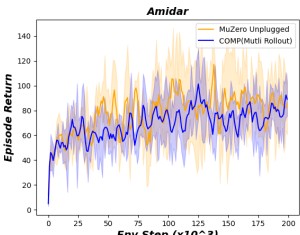

Figure 4: In these figures, we compare our algorithm with MuZero Unplugged in Atari games. We plot the IQM score, calculated over 3 seeds, for the MsPacman, Amidar, and Breakout tasks. Notably, in these comparative experiments, we add our regularization term to MuZero Unplugged and use similarity parameters for MCTS to ensure fairness.

## 4.3 Ablation Experiment

**Position of noise:** In Section 3.1, we analyze the error introduced by improper state translation and its influence on the model. Through this analysis, we discover that minimizing the error of the dynamics model and prediction model proves to be effective. Our experiments show whether other parts of the model also improve performance. We modify the regularization term of OneStep to compute the difference in policy, reward, value, and cosine similarity between the state and the noise state. Figure 5 left demonstrates that constraining cosine similarity and value difference can improve the algorithm's performance and validates our analysis. In addition, since the policy depends only

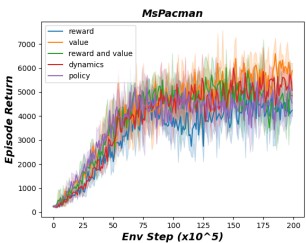 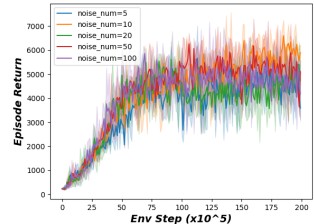 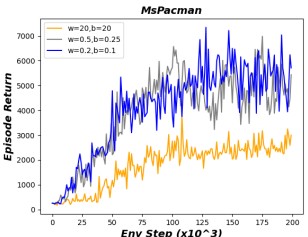

Figure 5: We demonstrate the performance of our algorithm under various conditions. On the left, we compare the impact of different regularization terms on various parts of the model. In the middle, we examine whether the number of noise instances influences the performance of our method. On the right, we illustrate the effect of different noise weights. The noise follows a specific distribution: $pw \sim \mathcal{N}(0, s * w + b)$

.

on the current state, constraining the policy yields little benefit. As for constraining the reward, it also offers little benefit in one-step rollouts, but in multi-step rollouts, it improves performance.

**Number of noise:** We wonder that whether the number of noise factors may influence our algorithm. Thus we set the number of noise factors to $n = 5, 10, 20, 50, 100$ to demonstrate this effect. As shown in the middle of Figure 5, $n = 10$ performs better and within a proper range, the algorithm maintains nearly consistent performance. This result relates to the analysis of Zheng et al. (2023). Its author points out that local Lipschitz conditions are sufficient. A strong regularization even can negatively impact the model. Therefore, we set this parameter as 10 in most environments.

**Weight of noise:** The weight of noise is an important parameter for our algorithm. We utilize noise in accordance with a distribution controlled by two parameters: $pw \sim \mathcal{N}(0, s * w + b)$. We set $w = 0.25, 0.5, 20.0$ and $b = 0.1, 0.25, 20.0$ respectively. Experiments demonstrate that within a proper range, the algorithm maintains nearly consistent performance. Notably, if the noise has an improper weight, the algorithm will fail. More discussion are listed in Appendix D.

## 5 RELATED WORK

Previous works, namely Levine et al. (2020) and Prudencio et al. (2023), provide a comprehensive survey of recent advancements in Offline RL research. These works categorize Offline RL methods based on various factors such as policy constraints, uncertainty estimation, and model-based approaches, among others. In this section, we will discuss some methods that are either compared or related to our algorithm.

Conservative Q-Learning (CQL Kumar et al. (2020)) is a regularization method that constrains the Q-value of out-of-dataset (OOD) state-action pairs to be lower than those of in-dataset state-action pairs. This algorithm belong to the category of model-free offline methods. In contrast, our algorithm falls under the model-based method category. Model-based offline reinforcement learning involves learning the dynamics function and reward function of the environment.

Model-based methods can be divided into two classes based on the method of action generation. One class includes algorithms that learn a world model to assist the agent in learning the policy, such as MOReL (Kidambi et al. (2020)), MOPO (Yu et al. (2020)), COMBO (Yu et al. (2021)), and others. These algorithms compute the uncertainty quantification of the transition state and process states with lower uncertainty. Although the theory and experiments of MOReL, MOPO, and COMBO have shown that their policies achieve notable results on state-based benchmarks such as D4RL, it remains unclear whether these algorithms are also effective in image-based domains such as Atari in the RL Unplugged benchmark. Additionally, there has been limited research on transferring these frameworks to image-based domains. The other class includes methods that use the world model to plan a future probability trajectory and select the action with the highest return, such as MuZero Unplugged (Schrittwieser et al. (2021)) and ROSMO (Liu et al. (2022)). These algorithms operate dynamics and prediction models on the latent state layer to enhance performance in image-based domains. Our algorithm falls into this category of algorithms.

## 6 CONCLUSION

In this paper, we analyze the improvement process of the value-based model-based planning. We identify latent errors originating from improper latent state translations during the evaluation phase. These errors propagate to the prediction model, generating misleading values. Consequently, an overestimated value impacts the improvement process by selecting a suboptimal action. Rather than reducing the rollout steps with ROSMO to avoid the accumulation of errors, we address this issue by introducing noise into the latent state. Through analysis and experimentation, we find that the dynamics and value components are more critical than others for value-based model-based planning. Therefore, we design two methods to separately introduce noise into the latent state for these two components. Experiments demonstrate that our algorithm effectively constrains the difference between the state and the noise state, thereby achieving state-of-the-art results.

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

## A   ALGORITHM PSEUDOCODE

We demonstrate the core procedure of COMP, which differs from ROSMO and MuZero Unplugged, as shown in the Algorithm 1 and 2.

---

**Algorithm 1** COMP (Constrained Dynamics)

---

**Require:** dataset $\mathcal{D}$, model parameters $\theta$, unroll step K, weight $\alpha$, init latent state $s_t^0$ of trajectory $\tau$, noise number n, noise wegith w
   **function** COSINE SIMILARITY($\theta$, $s_t^0$, $a_{t,...,t+K-1}$)
      $s \leftarrow Unroll(\theta, s_t^0, a_{t,...,t+K-1})$
      Sample n instances of noise with w from a normal distribution
      noise $s_t^{0*} \leftarrow s_t^0$ + noises
      $noises^* \leftarrow Unroll(\theta, s_t^{0*}, a_{t,...,t+K-1})$
      **for** $i = 1 \rightarrow n$ **do**
         Compute cosine similarity for s (stop gradient) with $s_i^*$
      **end for**
      **return** consine similarity
   **end function**

---

**Algorithm 2** COMP (Constrained Value)

---

**Require:** dataset $\mathcal{D}$, model parameters $\theta$, unroll step K, weight $\alpha$, latent state $s_t^0$ of trajectory $\tau$, noise number n, noise wegiht w
   **function** UNROLL($\theta$, $s_t^{0*}$, $a_{t,...,t+K-1}$)
      **for** $i = 0 \rightarrow K - 1$ **do**
         $\pi^i, v^i \leftarrow f_\theta(s^i)$
         $r^{i+1}, s^{i+1} \leftarrow g_\theta(s^i, a_i)$
         Sample n instances of noise with w from a normal distribution
         noise $s^{i*} \leftarrow s^i + noises$
         noise $\pi^{i*}, v^{i*} \leftarrow f_\theta(s^{i*})$
      **end for**
      $\pi^K, v^K \leftarrow f_\theta(s^K)$
      **return** $r, s, \pi, v, v^*$
   **end function**

---

For the loss functions of policy, value, and reward prediction, we employ the same functions as ROSMO and MuZero Unplugged. Specifically, the loss functions are:

$$
\begin{aligned}
\ell^p(\boldsymbol{\pi}, \mathbf{p}) &= -\mathbf{p}^\top \log \boldsymbol{\pi} \\
\ell^v(\mathbf{v}, z') &= -\phi(z')^\top \log \mathbf{v} \\
\ell^r(\mathbf{r}, u') &= -\phi(u')^\top \log \mathbf{r}
\end{aligned}
\tag{10}
$$

where we use the transform (rescale) function, $h(x) = \text{sign}(x)(\sqrt{|x| + 1} - 1 + \epsilon x)$, to obtain the invertible target value and reward, resulting in $z' = h(z), u' = h(r)$. For the transform function h, we set $\epsilon = 0.001$ in accordance with Pohlen et al. (2018). To compute the cross-entropy loss of the distribution predictions with the target, we apply the transformation $\phi$ to obtain the corresponding categorical representations of scalars.

## B   EXPERIMENT DETAILS

### B.1   EXPERIMENT SETTING

The behavior regularization for ROSMO is set at 0.2. The simulation budget for MuZero Unplugged is 20, with no limit on depth. For our algorithm, we sample noise from a normal distribution, apply a weight to it, and add it to the latent state to create a noise state. We have incorporated the official

| Parameter | Value |
|---|---|
| Frames stachked | 4 |
| Sticky action | True |
| Discount factor | $0.997^4$ |
| Batch size | 512 |
| Optimizer | Adamw |
| Optimizer learning rate | $7 \times 10^{-4}$ |
| Optimizer weight decay | $10^{-4}$ |
| Learning rate decay rate | 0.1 |
| Max gradient norm | 5 |
| Target network update interval | 200 |
| Policy loss coefficient | 1 |
| Value loss coefficient | 0.25 |
| Unroll length | 5 |
| TD steps | 5 |
| Bin size | 601 |

Table 1: Atari hyperparameters shared by MuZero Unplugged, ROSMO and COMP

| Environment | noise weight | consine similarity weight | noise value weight |
|---|---|---|---|
| Frostbite | 0.25 | 0.2 | 0.25 |
| Pong | 0.025 | 0.2 | 0.25 |
| Other Atari games | 1 | 0.2 | 0.25 |

Table 2: Atari hyperparameters of our algorithm. Different environments have varying tolerances for noise. Therefore, we select different noise weights and metrics as discussions in Appendix C.

MCTS library into ROSMO to implement MuZero Unplugged, adhering to the original parameter settings. The OneStep method, which is a baseline method of ROSMO, is implemented without behavior regularization. For all experiments conducted in Atari, we utilize 3 seeds. The comparison between the one-step and multi-step rollout methods with COMP involves merely the addition of a regularization term to the respective algorithm.

## B.2 NETWORK ARCHITECTURE

We utilize the same network architecture for MuZero Unplugged, ROSMO, and COMP in our Atari experiments. The representation, dynamics, and prediction functions in visual domains are modeled using ResNet v2 style pre-activation residual blocks with layer normalization.

We utilize the same network as introduced by ROSMO. Therefore, for the stacked grayscale image input of size 84 x 84 x 4, we first employ a downsampling network. Then, the representation function employs 6 residual blocks, while the dynamics function and prediction function utilize 2 residual blocks. All network blocks are kept consistent across different algorithms to ensure a similar neural network capacity. Additionally, all residual blocks have 64 hidden channels.

## B.3 PARAMETERS VALUE

We share a multitude of hyperparameters for ROSMO, MuZero Unplugged, and our algorithm, as shown in Table 1. Besides, hyperparameters specific to our algorithm are introduced in Table 2.

## C MORE EXPERIMENT RESULTS

Table 2 presents the IQM return results for various Atari games. In these experiments, we utilized the one-step rollout method of COMP. Our algorithm outperforms other algorithms in nearly all Atari games. In our algorithm, we found that the constraint value method performs better between

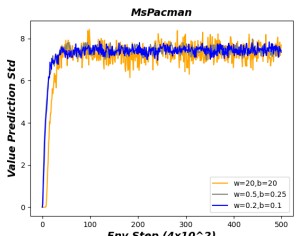 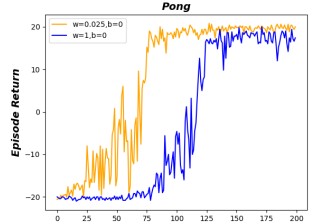 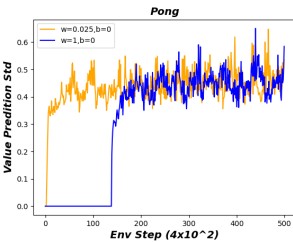

Figure 6: In these figures, we compare standard deviation of value of different noise weight. **left:** Standard deviation corresponding to the right figure in Figure 5. **middle:** Performance of different weight noise in Pong. **right:** Strandard devistion corresponding middel.

the two methods. This is because correcting errors in state translation is more complex than in value. Additionally, a smooth value tends to produce a more robust policy.

|  | CQL | MZU | ROSMO | COMP (dynamics) | COMP (value) |
|---|---|---|---|---|---|
| Amidar | $51.286_{\pm 18.821}$ | $76.452_{\pm 31.107}$ | $38.405_{\pm 9.501}$ | $67_{\pm 35.4}$ | $82.467_{\pm 45.867}$ |
| Asterix | $26890.476_{\pm 8496.429}$ | $29061.905_{\pm 3667.857}$ | $25740.476_{\pm 3857.143}$ | $25366.667_{\pm 2413.333}$ | $25813.333_{\pm 2946.667}$ |
| Breakout | $418.238_{\pm 27.571}$ | $390.119_{\pm 4.357}$ | $440.905_{\pm 5.774}$ | $408.6_{\pm 26.4}$ | $479.933_{\pm 49.733}$ |
| Frostbite | $3337.381_{\pm 544.643}$ | $4051.19_{\pm 107.5}$ | $3996.19_{\pm 223.006}$ | $3983.667_{\pm 118.667}$ | $4086.667_{\pm 188.667}$ |
| Gravitar | $52.381_{\pm 26.786}$ | $792.857_{\pm 108.929}$ | $753.571_{\pm 64.821}$ | $736.667_{\pm 136.667}$ | $776.667_{\pm 93.333}$ |
| MsPacman | $2141.667_{\pm 386.429}$ | $4539.048_{\pm 546.786}$ | $5019.762_{\pm 608.768}$ | $5072.667_{\pm 432.667}$ | $5892_{\pm 464}$ |
| Phoenix | $3510.238_{\pm 1066.429}$ | $6103.81_{\pm 1722.143}$ | $21550.476_{\pm 2689.643}$ | $22701.333_{\pm 2588.667}$ | $21358_{\pm 1432}$ |
| Pong | $18.762_{\pm 0.595}$ | $18.452_{\pm 1.107}$ | $20.452_{\pm 0.357}$ | $19.867_{\pm 0.267}$ | $20.267_{\pm 0.133}$ |
| Qbert | $13114.286_{\pm 797.321}$ | $13121.429_{\pm 933.929}$ | $15848.81_{\pm 1049.107}$ | $16758.333_{\pm 1678.333}$ | $16940_{\pm 915}$ |

Table 3: Numerical results of the IQM episode return of individual Atari games. The results of CQL, MUZ, ROSMO are sourced from Table 8 in the Liu et al. (2022)

## D    PARAMETER SELECT

### D.1    NOISE WEIGHT

The ablation experiment presented in Section 4.3 demonstrates that the weight of the noise has a significant impact on the performance of the algorithm. An improper range of noise weight can degrade the performance of the algorithm. We discuss how to identify an improper range of noise weight. One effective method to determine whether a weight is within a proper range is to observe the standard deviation of value prediction during the training phase. Figure 6 compares the standard deviation of value prediction for two noise methods, contrasting improper and proper weights. The figure on the left demonstrates that even if the standard deviation of value prediction varies slightly in the initial phase, it can still degrade performance. When comparing the performance of two weights in Pong, we discovered that the impact of an improper weight may be reduced, but it requires more steps. Furthermore, we can easily determine whether a weight is appropriate. Additionally, when comparing the left and right figures, we naturally conclude that the proper range may differ for different environments. This is related to the characteristics of the environment. Therefore, in a new environment, we can establish a standard by training with a smaller weight and shorter steps. For almost all Atari games, setting the weight to 1 is an appropriate value.

### D.2    NOISE NUMBER

Excluding the MsPacman game, we ran our algorithm with different noise numbers. As shown in Figure 7, in all environments, a noise number of 10 yields notable performance and better computational efficiency. Therefore, in all experiments, we set the noise number to 10.

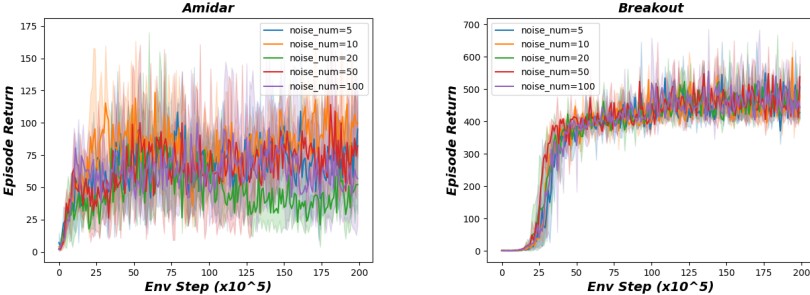

Figure 7: We utilize varying levels of noise to run other Atari games, demonstrating the impact of noise quantity.

