# OpenReview forum: "A PERSPECTIVE OF IMPROPER DYNAMICS ON OFFLINE MODEL-BASED PLANNING"
_ICLR.cc/2024/Conference — ICLR 2024 Conference Withdrawn Submission_

### Official Review · Reviewer_LURo · 2023-10-24

**Soundness:** 2 fair
**Presentation:** 2 fair
**Contribution:** 2 fair
**Rating:** 3
**Confidence:** 4

**Summary:**

The paper deals with model-based offline RL. It presents a modification of the algorithms MuZero Unplugged and ROSMO. It tries to increase the robustness by adding noise. The method is tested on several benchmarks and shows good results.

**Strengths:**

* The proposed modification can be used in various offline RL algorithms.
* The results are promising.

**Weaknesses:**

* The idea of using a special kind of noise to improve existing algorithms seems like a minor innovation.
* There is no discussion of the use of model uncertainty, which is also used to increase robustness.
* The focus on image-based benchmarks should appear much earlier in the text.
* The paper seems incomplete without the appendix. Both the pseudocode and the table of results can only be found in the appendix.

Further comments:

* Sentence "MuZero Unplugged (Schrittwieser et al. (2021)) and ROSMO (Liu et al. (2022)) are two offline reinforcement learning algorithms that generate actions by planning with a learned dynamics model" seems too vague to me. What does it mean to generate actions?

* In my opinion, "We add noise to the latent state to perform robust training" should also refer to the technique of using the uncertainty of the dynamics model in offline RL to increase robustness. See for example [1].

* "min" should not be italicized, see Eq. 8.

* "methods:(1)" -> "methods: (1)"

* In the discussion that starts with "Model-based methods can be divided into two classes" I recommend to extend the scope and also to call purely model-based methods that do not contain a Q- or value-function, see e.g. [2].

* The abbreviation IQM is used without having been introduced.

* In the references there are unintentional wrong lower case letters, e.g. monte-carlo, q-learning, rl, lipschitz

[1] Depeweg et al., Learning and policy search in stochastic dynamical systems with Bayesian neural networks, 2017

[2] Swazinna et al., Comparing model-free and model-based algorithms for offline reinforcement learning, 2022

**Questions:**

none

---

### Official Review · Reviewer_pj73 · 2023-10-26

**Soundness:** 2 fair
**Presentation:** 1 poor
**Contribution:** 1 poor
**Rating:** 1
**Confidence:** 4

**Summary:**

The describes a method (COMP) that injects noise in the latent state of MuZero and ROSMO. This is supposed to combat prediction errors of the transition model in offline model-based RL, and indeed the cosine similarity and a noised value loss are significantly improved in experiments on MsPacman. However, the performance of the method did not significantly improve over the baselines in MsPacman, Breakout and Amidar.

**Strengths:**

The paper addresses an important problem, offline model-based RL does need better transition models. The text also contains very few typos and the language is in general easy to follow.

**Weaknesses:**

The writing is often very hard to follow, as the authors often use terms without defining them properly. The reviewer was not able to follow the presented analysis in Section 3, which seems wrong. The presented method only seems to implement a known regularization technique and is not novel. Finally, while the experiments show a larger cosine-similarity and smaller noise value loss (which the noising should enforce), there does not seem to be a significant improvement in the returns over 3 Atari-games. In more detail:

1. Being familiar with the topic, the reviewer was able to guess what most of the terms in Section 2 were supposed to mean, but Section 3 is full of errors and unexplained variables. See the detailed comments for specifics.
2. The reviewer was unable to understand Section 3, but several key points seem to be wrong:
	- In (eq.7) the $Z(s)$ of $p^*$ is different from that of $p$ (different advantages lead to different normalization terms)
	- (Eq.8) is unclear (what is the minimum over), the last term should probably be a maximum and is expressed in terms of $s'$, which seems to be the "true" next state state, whereas $s'^*$ is the "wrong" next state. However, we don't know $s'$.
	- (Eq.9) seems wrong and also makes little sense to the reviewer: $a^{*k} - a^k$ is not an action that can result from an $\text{argmax}_a$, the argmax would yield the action where the two Q-values differ the most, but this action is not of any interest (only if the maximum action is different) and the final result should be an argmin.
3. The described algorithm seems to simply implement additional input noise, a technique well known in machine learning that regularizes functions to be smoother and is related to $L_2$ regularization. This technique might not have been applied to MuZero yet, but this combination is not very novel and would require an extraordinary effect to be of interest to ICLR.
4. Figure 2 shows a nice effect of COMP (except for the last plot, which does not show any real difference), but the error regions in the crucial Figures 3 and 4 overlap to an extend that it is impossible to say that COMP has any effect on the performance of both ROSMO and MuZero. Occam's razor demands that adding something to an algorithm needs to demonstrate significant improvement over the baseline, which does not seem to be the case here.

**Detailed Comments:**

- There are several terms that are not defined: plan function, prediction function, evaluation phase, discount marginal state-distribbution (which is never used), one-step process, observation, target network, translation error
- "and translating the latent state for the most probable action" seems wrong: in MuZero, the transition model translates the latent space for a given action
- (Eq.1): $\pi$ is not defined and $s_{t+1}$ is drawn from $T$, not $p$
- Clarify that $\pi_t^k$ is the prior policy, not the policy in (eq.1)
- A description of the planning process of MCTS is completely missing
- (Eq.2) uses a different font for the losses than the text
- "$p_{t+k}$ represents the improved action, as decided by the algorithm", it is never defined what the "improved action" is
- "regardless of whether the latent state can be accurately reconstructed in the input space", this should be the other way around, as the latent state is constructed from the observation space by the function $h_\theta$
- "value $^l_t$" misses a $v$
- "By repeating this process until the simulation budget is exhausted"; this is ambiguous: the process is not defined and one could think the simulation budget is equal to the depth
- (Eq.4) $T$ is undefined and the sums are missing things. Please also explain how $z_{MCTS}(s_t)$ and $z_{t+k}$ are related
- Please mention that (eq.5) is a cross-entropy loss
- It is not clear where $q_g(s,a)$ and $v(s)$ in (Eq.6) come from. Are they neural networks? How are they computed?
- In Section 3 it is not clear what the difference between $p^*$ and $p$ (and the follow ups) are
- What is the minimum in (Eq.8) over? The last equation only makes sense if the minimum becomes a maximum
- "the value of state-action pairs is still significant"; I think you mean "important"
- "when the model simulates a fixed number of steps." Why is the value unimportant if the number of steps goes to infinity? This and the next sentences characterize the working of MCTS wrong.
- "In this phase, the value of state-action pairs holds more importance than the visit count, leading us to overlook any errors in the visit count." This assertion needs to be justified
- "we discover that minimizing the error of action selection"; While I'm not sure what (eq.9) is not about, it is not about action selection
- "we add noise to all five steps of the unroll process"; what five steps? Is this the tree depth? The number of evaluated functions?
- Justify why you inject different noise for different measurements

**Questions:**

- What is the difference between COMP and a simple form of "data augmentation"?

---

### Official Review · Reviewer_dRm8 · 2023-10-28

**Soundness:** 2 fair
**Presentation:** 2 fair
**Contribution:** 2 fair
**Rating:** 3
**Confidence:** 4

**Summary:**

This paper introduces a method for enhancing offline model-based planning, by aligning the predicted value and/or dynamics of a latent state with those of a perturbed latent state. Experiments show improved performance in various Atari games compared to existing offline RL methods, including MuZero Unplugged, ROMSO and CQL.

**Strengths:**

1. The proposed method is well motivated - offline model-based RL algorithms often perform poorly when training data coverage is limited, due to the inaccuracy of their predictive models.
2. It is simple and compatible with some existing algorithms, such as MuZero Unplugged as illustrated in the paper, which offers the potential to improve existing frameworks.
3. Positive experimental results on a range of standard offline RL benchmarks.

**Weaknesses:**

1. The paper lacks of clarity. In section 3, it discusses two independent methods: constrained dynamics and constrained value, and proposes to use them to generate regularization terms for training the OneStep or MuZero Unplugged model. It's not clear to me what the exact objectives are. Writing the formulas out explicitly would help.
2. Most of the line plots are very hard to read, especially when the variance is high. It seems that most of the results are from 3 seeds. Would it make sense to increase the number of seeds to reduce the noise? Alternatively, it might be clearer to show the final performance of each baselines in a table.
3. The methods seem to be insensitive to its hyperparameters, which limits its applicability to different tasks.
4. The performance of the proposed method looks mixed compared to the existing baselines such as MuZero. For example, in table 3, both variants of COMP perform worse than MuZero on Asterix and Gravitar. Moreover the relative performance between the two variants also vary from task to task.

**Questions:**

1. In 3.3, you mention that you use two types of noise. Which one did you use for each experiment? How do their performance vary in different tasks? What are the hyperparameters used, e.g. sampling weights?
2. What are the exact regularization terms for "constrained dynamcis" and "constrained value"?
3. Could you show the aggregated performance on all atari tasks?

---

### Official Review · Reviewer_NRcG · 2023-10-31

**Soundness:** 2 fair
**Presentation:** 2 fair
**Contribution:** 1 poor
**Rating:** 3
**Confidence:** 4

**Summary:**

This paper proposes to regularize the dynamics model learning by minimizing the  difference between the predicted value/states and their noisy counterparts. The experiments are performed on several Atari games, compared with baselines ROSMO and MuZero Unplugged, and one simplified version of ROSMO, Onestep.

**Strengths:**

The problem of handling inaccurate dynamics model in planning is important;

**Weaknesses:**

- The proposed method of regularizing the training by sampled noise is straightforward and lacks significant novelty. It can be seen as a practical implementation trick. Of course, with intensive benchmarking experiments, such tricks can also be published in a prestigious venue. However, this paper doesn’t go that way due to the following points.
- The experiments are not sufficient. Many recent model-based methods are proposed but are not compared. Including more complex benchmarks such as D4RL could be more convincing.
- The performance improvement is modest. I don’t see significant improvement compared to the baselines.
- The writing needs to be heavily polished.

**Questions:**

- What is OneStep?
- What does it mean by “state translation”? Do you mean state transition?
- Why is Equation (8) valid? How is the last step derived?